# *Escherichia coli* adhesion portion FimH functions as an adjuvant for cancer immunotherapy

Wei Zhang [1,6], Li Xu [1,6], Hae-Bin Park[1,2], Juyoung Hwang[1,2], Minseok Kwak[3], Peter C.W. Lee [4], Guang Liang[5], Xiaoyan Zhang[1], Jianqing Xu [1] & Jun-O Jin [1,2✉]

Induction of antigen-specific immune activation by the maturation of dendritic cells (DCs) is a strategy used for cancer immunotherapy. In this study, we find that FimH, which is an *Escherichia coli* adhesion portion, induces toll-like receptor 4-dependent and myeloid differentiation protein 2-independent DC maturation in mice in vivo. A combined treatment regimen with FimH and antigen promotes antigen-specific immune activation, including proliferation of T cells, production of IFN-γ and TNF-α, and infiltration of effector T cells into tumors, which consequently inhibits tumor growth in mice in vivo against melanoma and carcinoma. In addition, combined therapeutic treatment of anti-PD-L1 antibodies and FimH treatment efficiently inhibits CT26 tumor growth in BALB/c mice. Finally, FimH promotes human peripheral blood DC activation and syngeneic T-cell proliferation and activation. Taken together, these findings demonstrate that FimH can be a useful adjuvant for cancer immunotherapy.

[1] Shanghai Public Health Clinical Center & Institutes of Biomedical Sciences, Fudan University, Shanghai 201508, China. [2] Department of Medical Biotechnology, Yeungnam University, Gyeongsan 38541, South Korea. [3] Department of Chemistry, Pukyong National University, Busan 48513, South Korea. [4] Department of Biomedical Sciences, University of Ulsan College of Medicine, ASAN Medical Center, Seoul 05505, South Korea. [5] Chemical Biology Research Center, School of Pharmaceutical Sciences, Wenzhou Medical University, Wenzhou, Zhejiang 325035, China. [6] These authors contributed equally: Wei Zhang, Li Xu. ✉email: jinjo@yu.ac.kr

mmunotherapy is an emerging treatment method for cancer, and includes approaches, such as immune checkpoint blockade antibody (Ab) treatment, chimeric antigen (Ag) receptor T-cell therapy, and cancer vaccines[1–3]. Although many novel immunotherapy methods are being developed, a combined treatment method of adjuvants and cancer antigens is desirable as a form of traditional immunotherapy. This is because a combined adjuvant and Ag treatment reduces unwanted inflammation through Ag-specific immune responses and can be developed as a cancer vaccine[3,4]. A combination therapy of the immune checkpoint blockade and immune activation promotion by stimuli is also being studied as a form of advanced immunotherapy[5–7].

Since cancer antigens are poorly immunogenic, adjuvants are often added to treatments to enhance the activation of immune responses[8]. The stimulation of pattern-recognition receptors (PRRs) in antigen-presenting cells (APCs) induces innate and adaptive immune activation. Toll-like receptors (TLRs) are well-studied PRRs in APCs, such as dendritic cells (DCs) and macrophages[9]. Lipopolysaccharide (LPS) is a TLR4 ligand and induces strong immune activation[10]. However, LPS is an endotoxin that promotes a cellular stress response and superoxide production, inducing fatal sepsis syndrome in humans and animals. To reduce the endotoxic effect of LPS, it has been modified to monophosphoryl lipid A (MPLA)[11]. An adjuvant system using MPLA is licensed and underwent phase-III clinical trials for several vaccines[12–15]. However, MPLA alone is not water-soluble and requires formulation with aluminum salt precipitates for the enhancement of immune activation[16]. Nevertheless, MPLA still retains its toxic effect[12,17]. Therefore, it is necessary to study other TLR4 stimulants that are water-soluble and less toxic.

For effective immunotherapy against cancer, cancer-Ag-specific immune activation is required to efficiently kill cancer cells[4,8]. DCs are powerful APCs for the promoting Ag-specific adaptive immune activation, including the activation of CD4+ helper T (Th) cells and CD8+ cytotoxic T lymphocytes (CTL)[18,19]. The stimulation of DCs by an immunogenic Ag or adjuvant induces the upregulation of costimulatory molecules, pro-inflammatory cytokine production, and Ag presentation[18]. Depending on the phenotype, the role of DCs is distinguished by the presentation of the Ag. CD8α+ DCs in mice and BDCA3+ DCs in humans efficiently promote cross-presentation of the Ag of the major histocompatibility complex (MHC) class I to CD8 T cells. In contrast, CD8α− DCs in mice and BDCA1+ DCs in humans present the Ag of MHC class II directly to CD4 T cells[20–22]. The presentation of the Ag by these DCs is essential for inducing Ag-specific immune activation for cancer immunotherapy and vaccines.

FimH is the adhesion portion of type 1 fimbriae in *Escherichia coli* (*E. coli*)[23]. As enteropathogenic *E. coli* (EPEC) and enterohemorrhagic *E. coli* (EHEC) attach to mucosal epithelial surfaces and lead to severe diarrhea, vomiting, and fever with high rates of fatality, FimH is used to protect protein molecules from intestinal infection of EPEC and EHEC[24]. FimH has been studied as an inducer of natural killer (NK)-cell activation via TLR4 stimulation[25]. However, the adjuvant effect of FimH for cancer immunotherapy, especially for DC-mediated Ag-specific immune activation and enhancement of anti-PD-L1 effect in immunotherapy, has not been investigated. Since FimH stimulates TLR4, we hypothesize that it may function as an adjuvant and promote DC-mediated Ag-specific immune activation.

In this study, we evaluate whether FimH functions as an adjuvant for cancer treatment by immunotherapy. The FimH from *E. coli* and yeast promotes spleen and lymph node DC activation, and the combination treatment of FimH and Ag induces Ag-specific immune activation, which consequently inhibits Ag-expressing tumor growth in mice in vivo. Moreover,

FimH enhances the anti-PD-L1-induced anti-cancer effect. Therefore, these data suggest that FimH functions as an adjuvant for enhancing immune responses against cancer.

## Results

**TLR4-dependent activation of DCs by FimH in mice in vivo.** As FimH promotes the activation of innate immune cells in vitro and in vivo[23,25], we examined whether it can induce the activation of lymph node (LN) DCs in mice in vivo. FimH was purified from *E. coli*, and the endotoxin was removed by a column treatment method, as detailed in the "Methods" section (Supplementary Fig. 1). C57BL/6 mice were injected subcutaneously (s.c.) with 2.5 mg per kg of FimH and 1 mg per kg of LPS. The DC subsets were defined from the lineage− CD11c+ cells in DAPI− live cells (Supplementary Fig. 2). Twenty-four hours after injection, expression levels of costimulatory molecules and MHC class I and II were significantly upregulated by FimH treatment in CD8α+ and CD8α− DCs in the inguinal LN (iLN) (Fig. 1a). Moreover, the activation of DCs was dose-dependently increased by FimH, of which 2.5 and 5 mg per kg of FimH promoted higher expression levels of MHC class I and II compared to those induced by LPS (Supplementary Fig. 3). Furthermore, the upregulated costimulatory molecules and MHC class I and II levels by FimH were maintained for much longer than the LPS-induced the levels (Supplementary Fig. 4). Consistent with previous studies showing that FimH promotes TLR4-dependent activation in NK cells[25], the upregulation of the expression of the costimulatory molecule by FimH in iLN DCs was completely eliminated in the TLR4-knockout (KO) mice (Fig. 1b). In addition, treatment with FimH promoted substantial enhancement of the pro-inflammatory cytokine production in the serum (Fig. 1c). Moreover, treatment with FimH in the mice induced a significant upregulation of IFN-γ and T-bet mRNA levels-critical transcription factors for helper T1 (Th1) and cytotoxic T1 (Tc1) cells, in the iLN (Fig. 1d). However, Th2- and Th17-related mRNA levels were not upregulated by FimH (Supplementary Fig. 5). Consistent with the expression of the costimulatory molecule in TLR4-KO mice, the pro-inflammatory cytokine levels in the serum and the IFN-γ and T-bet mRNA levels in the iLN were not upregulated by FimH in the TLR4-KO mice (Fig. 1c, d). Thus, these data indicate that FimH promotes TLR4-dependent DC activation.

Since LPS is a well-known immunostimulatory molecule and can be easily contaminated in a laboratory, we purified FimH from yeast by fermentation and examined the capacity of DC activation in the mouse (Supplementary Fig. 6a). The treatment of yeast-purified FimH promoted substantial upregulation of costimulatory molecules and MHC class I and II, which was almost similar to *E. coli*-isolated FimH (Supplementary Fig. 6b). Thus, these findings indicated that the effect of FimH in the DC activation was not dependent on the contamination of LPS.

**Comparison of FimH and LPS toxicity.** Although LPS shows a powerful immune-stimulatory effect, it cannot be used for humans as an adjuvant due to its toxicity. Therefore, we examined whether FimH also promotes toxicity in mice. The FimH treatment did not induce IL-1β production, which indicates activation of inflammasomes in the cells, in the bone marrow-derived DCs (BMDCs) with or without ATP (an enhancer of IL-1β production) (Fig. 2a). Moreover, levels of serum pro-inflammatory cytokines (the main toxic molecules produced due to immune stimulation) induced by high concentration of FimH were much lower than those induced by 20 mg per kg of LPS (Fig. 2b). Furthermore, 70% of the mice that were administered intravenous (i.v.) injection of FimH at a dose of 250 mg per kg, survived for longer than 84 h; however, the

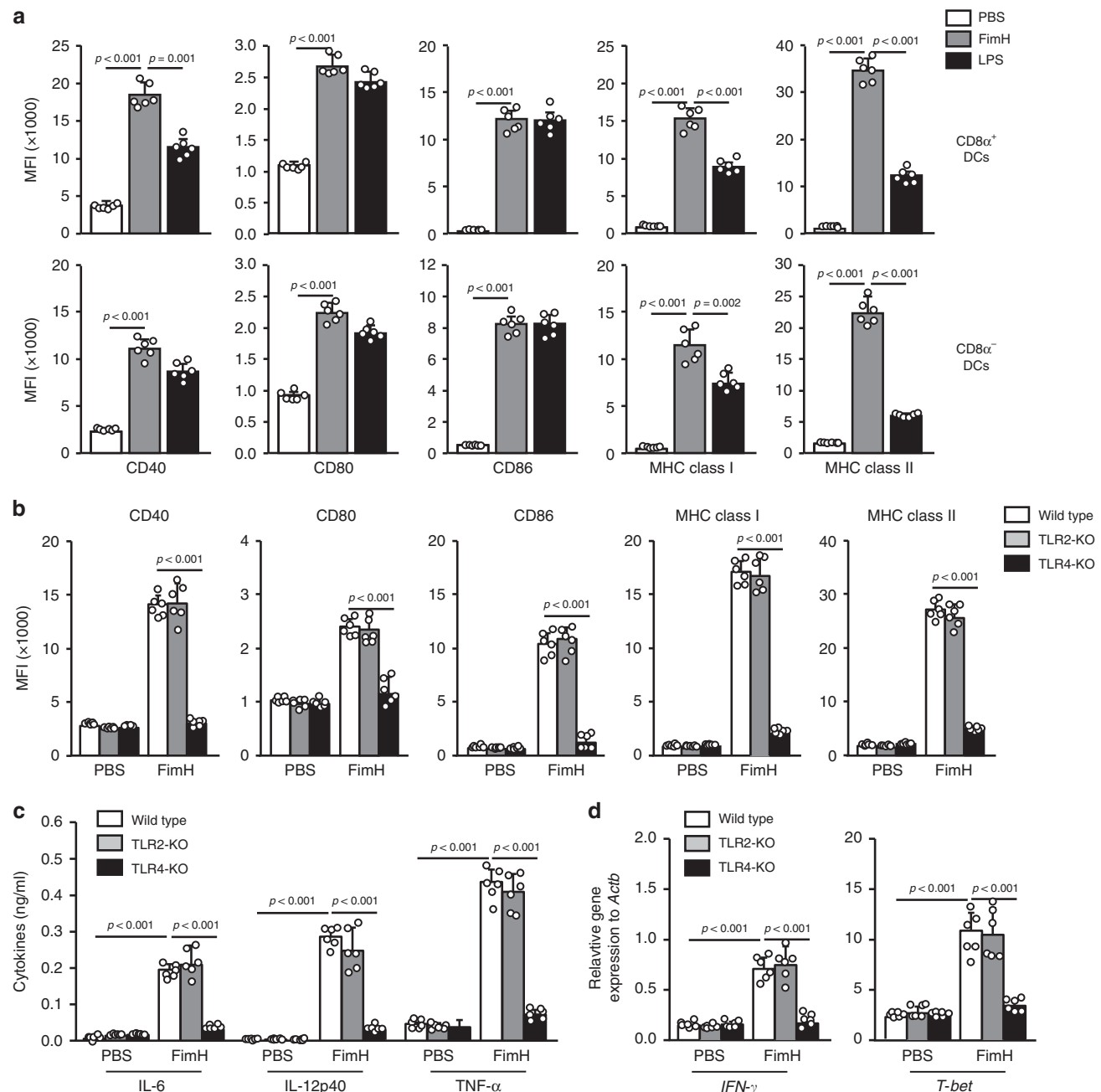

**Fig. 1 FimH promotes the activation of DCs in inguinal lymph nodes (iLN) in mice in vivo.** C57BL/6 mice were injected subcutaneously (s.c.) with 2.5 mg per kg of FimH and 1 mg per kg of LPS for 24 h, and the inguinal lymph nodes (iLN) were harvested. **a** The expression levels of costimulatory molecules and MHC classes I and II in CD8α⁺ DCs (upper panel) and CD8α⁻ DCs (lower panel) are shown. ($n = 6$ mice, one-way ANOVA, mean ± SEM). **b** Flow cytometry analysis of costimulatory molecules and MHC classes I and II in iLN DCs from toll-like receptor 2 (TLR2)- and TLR4-knockout (KO) mice. ($n = 6$ mice, two-way ANOVA, mean ± SEM). **c** Serum concentrations of IL-6, IL-12p40, and TNF-α in wild type, TLR2-KO and TLR4-KO mice are shown. ($n = 6$ mice, two-way ANOVA, mean ± SEM). **d** Mean values of the mRNA expression levels in the iLN are shown. ($n = 6$ mice, two-way ANOVA, mean ± SEM).

mice treated with 20 mg per kg of LPS died within 36 h (Fig. 2c). The treatment of 250 mg per kg of FimH did not promote inflammation in the lungs (Fig. 2d). Thus, these findings suggest that FimH has much lower cytotoxicity in mice than does LPS.

The activation of TLR4 by LPS requires myeloid differentiation protein 2 (MD2). For evaluation of the different effect of FimH compared to LPS, we treated FimH and LPS in MD2-KO mice and found that there is FimH–induced upregulation of costimulatory molecules and MHC class I and II expression, while LPS did not promote upregulation of these molecules in MD2-KO

mice (Fig. 2e). Thus, these data indicated that FimH-induced activation of DC is independent of MD2.

**FimH induces Ag-specific Th1 and Tc1 activation.** Since FimH had induced DC and T-cell activation in the iLN, we next assessed whether it could promote Ag-specific immune activation in tumor-bearing mice in vivo. To determine the Ag-specific T-cell activation in tumor-bearing mice, CFSE-labeled OT-I and OT-II cells were transferred into B16-OVA-bearing CD45.1

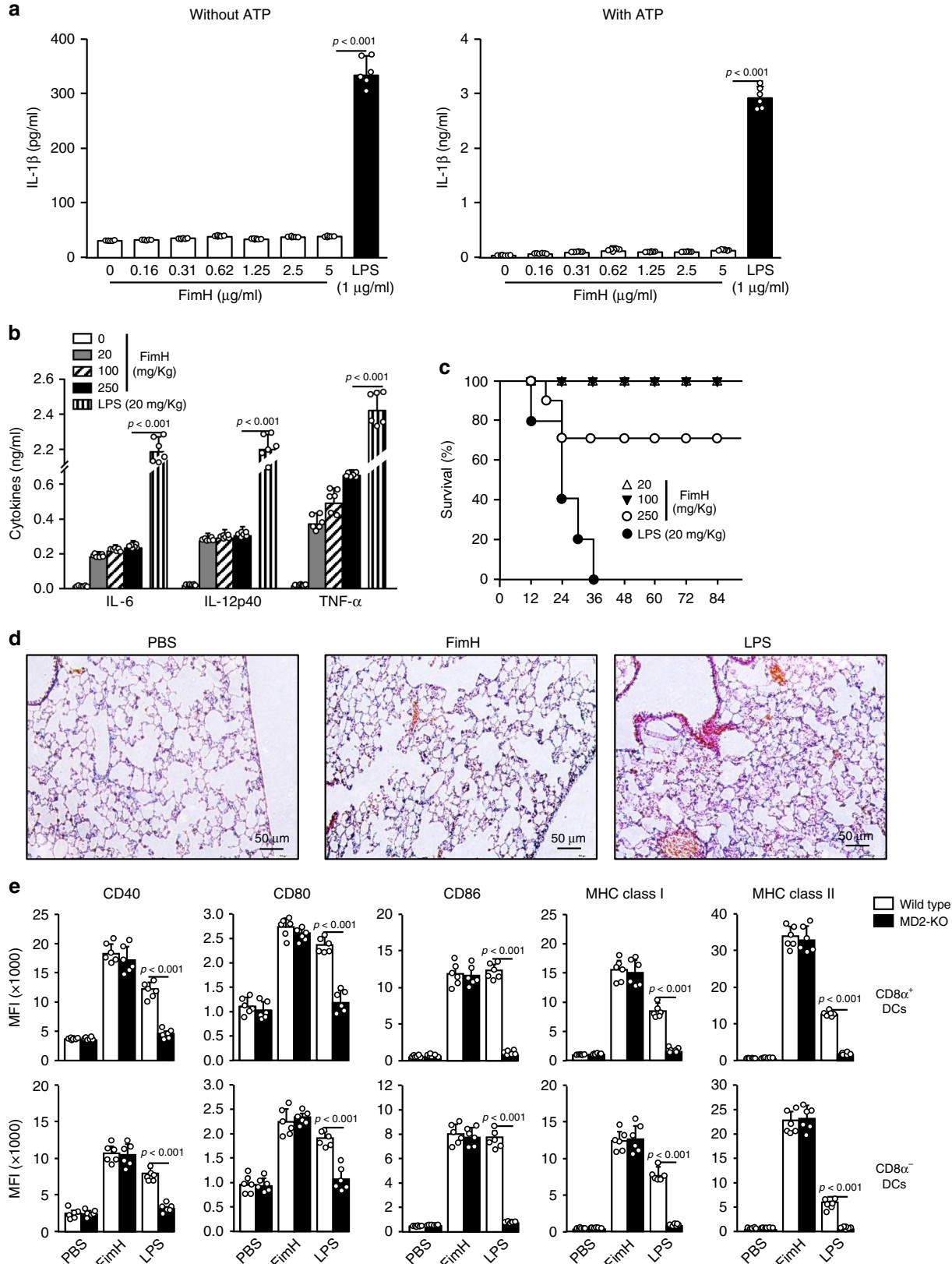

congenic mice. After 24 h, the mice received 2.5 mg per kg of OVA, 2.5 mg per kg of FimH, or a combination of 2.5 mg per kg OVA and 2.5 mg per kg FimH in PBS. Another group was also treated with 2.5 mg per kg of OVA and 1 mg per kg of LPS as a positive control. Four days after transfer of cells, the tumor and

tumor-draining lymph nodes (drLN) were harvested and analyzed OT-I and OT-II cell proliferation (Supplementary Fig. 7). We observed that the combined treatment with OVA and FimH had induced a marked increase in the proliferation of OT-I and OT-II cells in the tumor-drLN, which was significantly

**Fig. 2 FimH induces relatively low cytotoxicity in mice than LPS. a** Bone marrow-derived DCs (BMDCs) were treated with the indicated doses of FimH and LPS in the presence and absence of ATP. IL-1β production levels in cultured medium was measured 24 h after treatment. *P*-value were defined by the unpaired *t*-test ($n = 6$ mice, one-way ANOVA, mean ± SEM). **b** IL-6, IL-12p40, and TNF-α production levels in serum were analyzed by ELISA 24 h after injection of FimH and LPS. The unpaired *t*-test were used. ($n = 6$ mice, one-way ANOVA, mean ± SEM). **c** Survival rates of mice after intravenous treatment of FimH and LPS ($n = 10$ in each group). **d** Hematoxylin and eosin (H&E) of lungs are shown 24 h after treatment. **e** C57BL/6 mice and myeloid differentiation protein 2 (MD2)-KO mice were injected s.c. with 2.5 mg per kg of FimH and 1 mg per kg of LPS for 24 h, and splenocytes were harvested. Flow cytometry analysis of costimulatory molecules and MHC classes I and II in CD8α+ DCs (upper panel) and CD8α− DCs (lower panel) are shown. ($n = 6$ mice, two-way ANOVA, mean ± SEM).

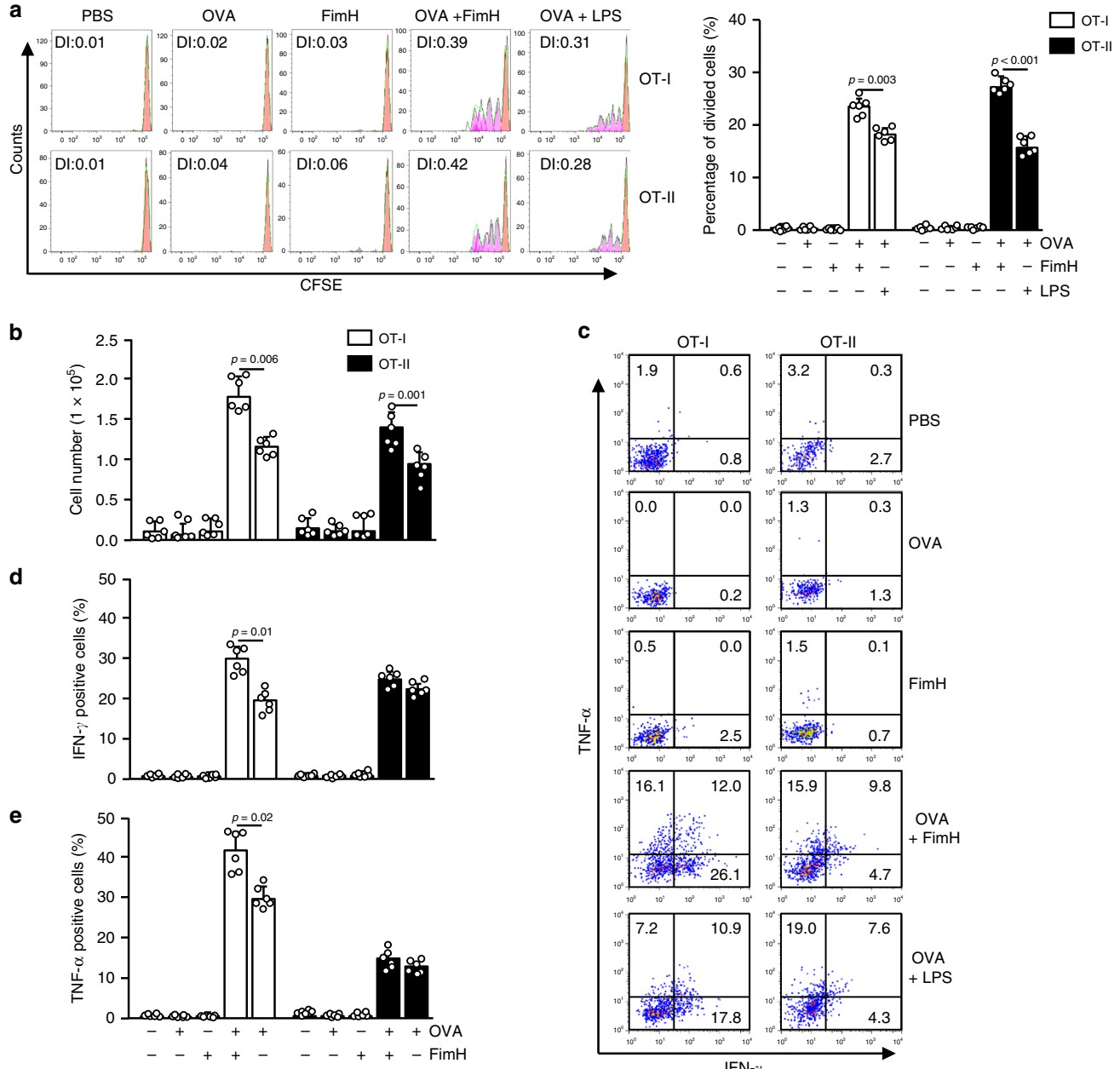

**Fig. 3 FimH enhances OVA-specific T-cell immunity in mice in vivo.** C57BL/6 mice were injected *s.c.* with $1 \times 10^6$ B16-OVA cells. Ten days after tumor-cell injection, CFSE-labeled OT-I and OT-II cells were transferred into the mice and subsequently treated with 2.5 mg per kg of OVA, 2.5 mg per kg of FimH, or a combination of OVA and FimH for 24 h. Tumor-draining lymph nodes (drLN) and tumors were harvested 3 days after treatment. **a** The proliferation of OT-I and OT-II T cells in the tumor-drLN of the CD45.1 congenic mice (left panel) and the mean percentages of the proliferating cells are shown (right panel). Division index (DI) was calculated by proliferation assay tool in flowjo software. ($n = 6$ mice, one-way ANOVA, mean ± SEM). **b** The absolute numbers of the tumor-infiltrating OT-I and OT-II cells. ($n = 6$ mice, one-way ANOVA, mean ± SEM). **c** The percentages of IFN-γ- and TNF-α-producing cells within the tumor-infiltrated OT-I and OT-II cells. **d** The mean percentages of IFN-γ- and **e** TNF-α-producing OT-I and OT-II cells are shown. ($n = 6$ mice, one-way ANOVA, mean ± SEM).

higher than that of the LPS-induced cells (Fig. 3a). The tumor-infiltrating OT-I and OT-II cells were also substantially increased by the combined treatment of FimH and OVA (Fig. 3b). In addition, the combined treatment of FimH and OVA markedly increased intracellular IFN-γ- and TNF-α-producing OT-I and OT-II cells in the B16-OVA tumor (Fig. 3c–e). These data suggest that FimH promotes Ag-specific T-cell activation in tumor-bearing mice together with the proliferation of these cells and also IFN-γ and TNF-α production.

**Anti-OVA effect through the combination of OVA and FimH.** Our data indicated that the combined FimH and OVA treatment promoted OVA-specific immune activation; therefore, we examined its anti-tumor effect against a B16-OVA tumor. C57BL/6 mice were injected s.c. with $1 \times 10^6$ B16-OVA cells. Seven days after tumor-cell

injection, the mice were treated s.c. with PBS, 2.5 mg per kg of OVA, 2.5 mg per kg of FimH, or a combination of OVA and FimH. LPS combined with OVA were also administered to tumor-bearing mice as a positive control. Fourteen days after cell injection, the mice received the same treatments for a second round. As shown in Fig. 4a, b, the combined treatment with OVA and FimH effectively prevented B16-OVA tumor growth. In addition, 21 days after the tumor-cell injection, the mice treated with the combination of FimH and OVA showed substantially greater OVA-specific IFN-γ production in the tumor-drLN cells by the re-stimulation of the OVA(257–264) and (323–339) peptides than by other treatments (Fig. 4c). Furthermore, the combined treatment with FimH and OVA induced substantially greater OVA-specific cytotoxicity against OVA peptide-pulsed target cells in mouse spleens than those treated with other controls (Fig. 4d). Interestingly, the Ag-specific immune responses and anti-cancer effects induced by the

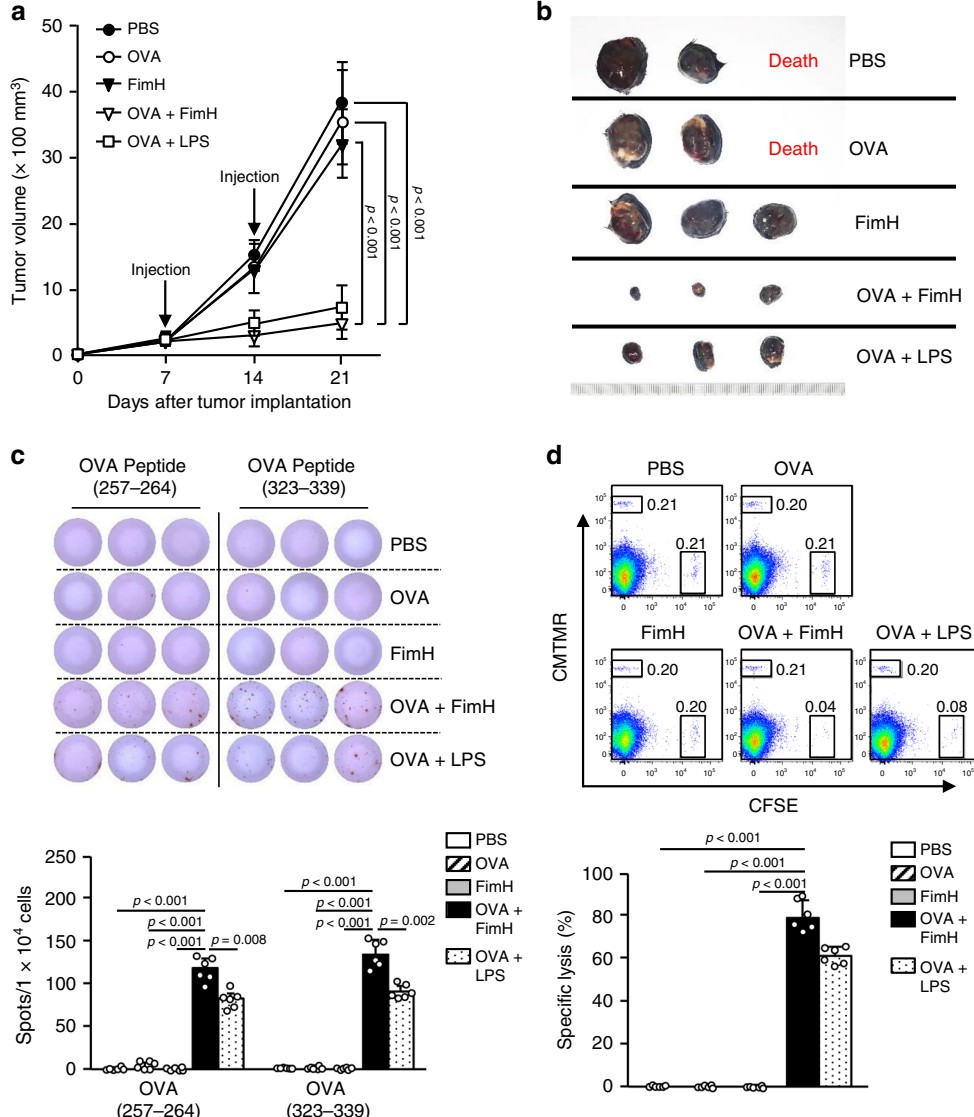

**Fig. 4 Combined OVA and FimH treatment inhibits B16-OVA tumor growth.** C57BL/6 mice were injected s.c. with $1 \times 10^6$ B16-OVA cells on the right side. After 7 and 14 of tumor-cell injection, the mice received s.c., 2.5 mg per kg of OVA, 2.5 mg per kg of FimH, or a combination of OVA and FimH in PBS. **a** The B16-OVA tumor growth curves are shown. ($n = 6$ mice, significance determined by the log-rank test, mean ± SEM. **b** Representative tumor masses on day 21 are shown. ($n = 6$ mice). **c** OVA peptides (257–264) and (323–339) specific IFN-γ production in iLN cells were measured by ELISpot analysis (upper panel). The mean numbers of spots are shown (lower panel). ($n = 6$ mice, one-way ANOVA, mean ± SEM). **d** Ag-specific cytotoxic activity was measured in the mouse in vivo. Dot plots showed the percentages of SIINFEK-loaded CFSE$^+$ cells and CMTMR$^+$ cells not loaded with peptide (upper panel). The mean percentages of Ag-specific lysis are shown (lower panel). ($n = 6$ mice, one-way ANOVA, mean ± SEM).

combination of OVA and FimH were considerably higher than the effects induced by the combination of OVA and LPS. Thus, these data suggest that the combined Ag and FimH treatment induced an anti-cancer effect in mice in vivo via the activation of an Ag-specific immune responses.

**Cancer self-Ag- and FimH-induced anti-cancer immunity.** The combination of OVA and FimH promoted OVA-specific immune responses and anti-cancer effects against B16-OVA cells; therefore, we next evaluated the anti-cancer effects of FimH and self-antigens of carcinoma and melanoma in BALB/c and C57BL/6 mice, respectively. The BALB/c mice were injected s.c. with $1 \times 10^6$ CT26 cells and the C57BL/6 mice with $1 \times 10^6$ B16 cells. After 7 and 14 days of tumor-cell injections, the BALB/c mice were treated s.c. with PBS and 2.5 mg per kg of AH1A5, 2.5 mg per kg

of FimH, or the combination of FimH and AH1A5. The C57BL/6 mice received a combination of FimH and tyrosinase-related protein 2 (TRP2, a self-Ag of melanoma) and other controls. The combined treatments with FimH and the self-Ags efficiently reduced the growth of the carcinoma and melanoma in the BALB/c and C57BL/6 mice, respectively (Fig. 5a, b). As shown in Fig. 5c, d, the sizes of the tumor masses after the combined treatments were also noticeably smaller than those in the other controls. Twenty-one days after the tumor-cell injection, the self-Ag-specific IFN-γ production in the tumor-drLN cells after the combined treatment of self-Ag and FimH was significantly higher than it had been with the other treatments (Fig. 5e, f). Consistent with our findings on the OVA-specific immune activation by FimH and OVA treatment, the self-Ag-specific and anti-cancer effects of FimH and the self-Ag were considerably greater than

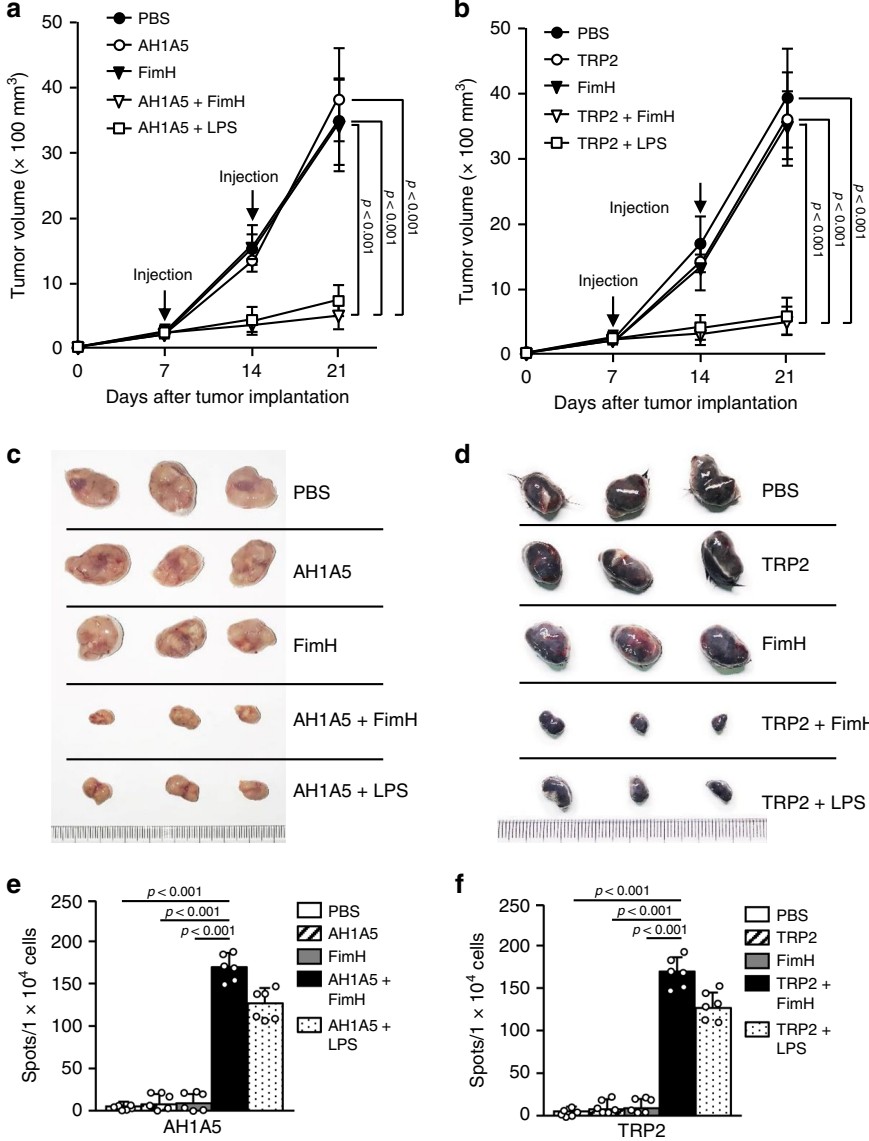

**Fig. 5 FimH promotes cancer antigen (Ag)-specific immune activation and anti-tumor immunity.** BALB/c mice were injected s.c. with $1 \times 10^6$ CT26 carcinoma cells. After 7 and 14 days of tumor-cell injections, the mice were treated with PBS and 2.5 mg per kg of AH1A5, 2.5 mg per kg of FimH, or a combination of AH1A5 and FimH. The C57BL/6 mice were injected s.c. with $1 \times 10^6$ B16 melanoma cells. After 7 and 14 days of tumor-cell injections, the mice received PBS and 2.5 mg per kg of TRP2, 2.5 mg per kg of FimH, or a combination of TRP2 and FimH. **a** The growth curves of the CT26 tumors in the BALB/c mice and **b** the B16 tumors in the C57BL/6 mice are shown. ($n = 6$ mice, significance determined by the log-rank test, mean ± SEM. **c** The CT26 carcinoma and **d** representative B16 melanoma tumor masses are shown. ($n = 6$ mice). **e** AH1A5-specific and **f** TRP2-specific IFN-γ production in the tumor drLN of the BALB/c and C57BL/6 mice are demonstrated, respectively. ($n = 6$ mice, one-way ANOVA, mean ± SEM).

those induced by the combined LPS and Ag treatment. Thus, these data suggest that FimH can induce self-Ag-specific immune activation and inhibit tumor growth in mice in vivo.

**Mucosal adjuvant effect of FimH**. We next evaluated the mucosal adjuvant effect of FimH. C57BL/6 mice were treated intranasally (i.n.) with 2.5 mg per kg of FimH and 1 mg per kg of LPS. Twenty-four hours after treatment, the mediastinal LN (mLN) were harvested and analyzed for DC activation (Supplementary Fig. 8). The costimulatory molecules and MHC class I and II expression levels were found to be markedly increased by FimH treatment (Fig. 6a). Moreover, the proliferation of the

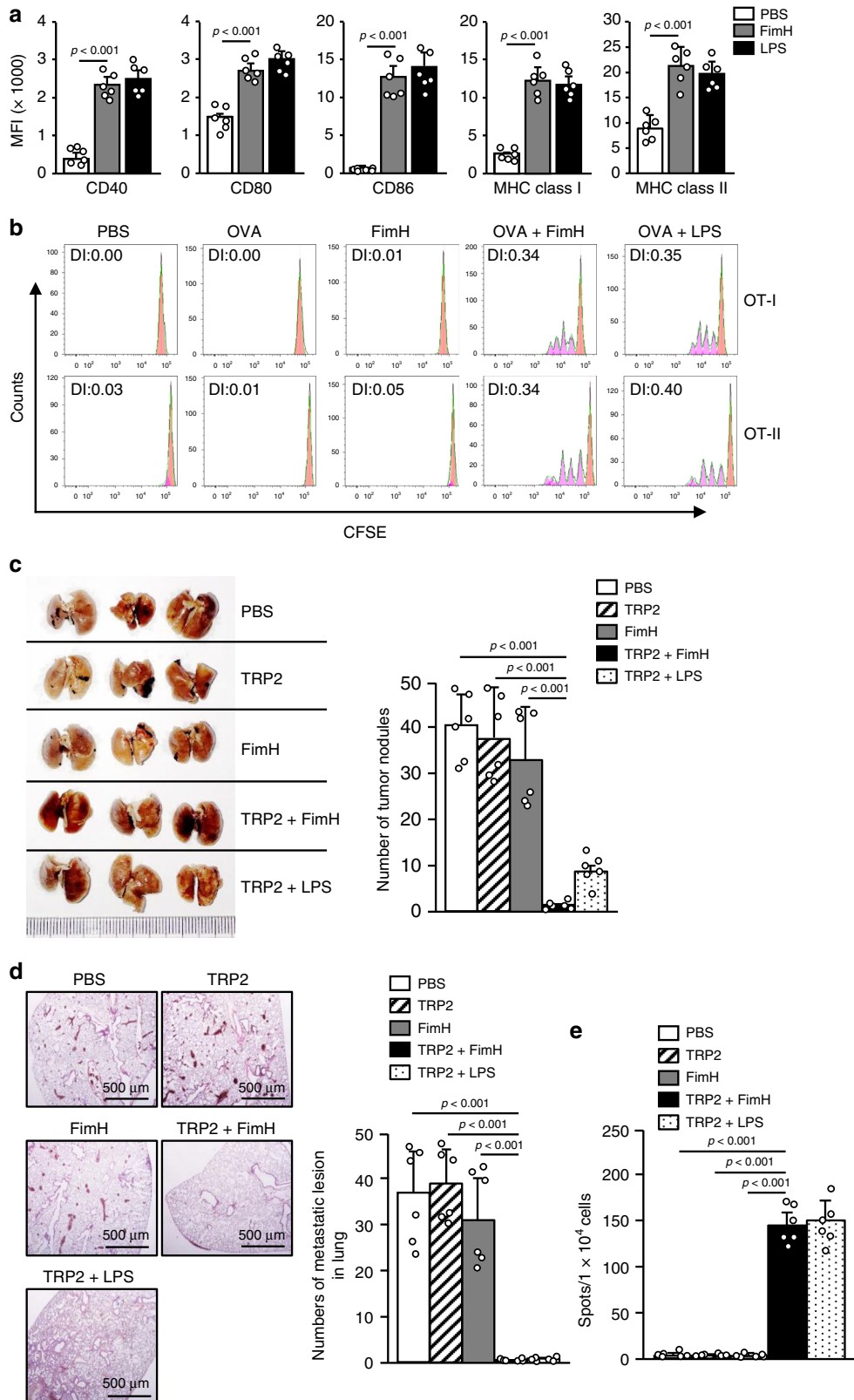

**Fig. 6 FimH functions as a mucosal adjuvant for the inhibition of lung metastasis of melanoma.** C57BL/6 mice were injected intranasally (i.n.) with 2.5 mg per kg of FimH or 1 mg per kg of LPS. **a** The mean fluorescence intensity (MFI) levels of costimulatory molecules and MHC classes I and II in the mediastinal-LN (mLN) DCs are shown 24 h after treatment. (n = 6 mice, one-way ANOVA, mean ± SEM). **b** Transferred OT-I and OT-II cell proliferation was measured by flow cytometry. **c–e** C57BL/6 mice were injected intravenously (i.v.) with 1 × 10$^6$ B16 melanoma cells and treated i.n. with PBS and 2.5 mg per kg of TRP2, 2.5 mg per kg of FimH, or a combination of TRP2 and FimH after 5 and 10 days of the tumor-cell injections. **c** Metastatic B16 tumor cells in the lungs after 14 days of the tumor-cell injection (left panel) and number of surface tumor nodules are shown (right panel). (n = 6 mice, one-way ANOVA, mean ± SEM). **d** H&E stained lung sections (left panel) and numbers of metastatic lesions in the lung (right panel) after 14 days of tumor-cell injection are shown. **e** TRP2-specific IFN-γ production in the mLN cells was analyzed by ELISpot analysis. (n = 6 mice, one-way ANOVA, mean ± SEM).

CFSE-labeled and transferred OT-I and OT-II cells was substantially increased in the mLN by the i.n. combined treatment with FimH and OVA (Fig. 6b). In addition, the combined i.n. treatment of mice with FimH and TRP2 effectively prevented melanoma tumor growth in the lungs (Fig. 6c, d). The mLN cells of mice treated i.n. with the combination of FimH and TRP2 also produced, by re-stimulation of the TRP2 peptide, higher amounts of TRP2-specific IFN-γ than did those with other treatments (Fig. 6e). Therefore, these data suggest that FimH functions as a mucosal adjuvant for lung cancer immunotherapy.

**Enhancement of anti-PD-L1 antibody effect by FimH.** Next, we examined the effect of FimH on the enhancement of the anti-cancer effects that were induced by the immune check point blockade antibodies (Abs). CT26 tumor-bearing BALB/c mice were treated intraperitoneally (i.p.) with 10 mg per kg anti-PD-L1 Abs, 2.5 mg per kg FimH, and a combination of anti-PD-L1 Abs and FimH every 5 days until 25 days after tumor injection. A combination of 1 mg per kg LPS and anti-PD-L1 Abs had then been administered to tumor-bearing mice as a positive control. The treatment of anti-PD-L1 Abs efficiently inhibited CT26 tumor growth; in addition, the tumor was completely abrogated by the combination of anti-PD-L1 Abs and FimH (Fig. 7a). Moreover, the TNF-α and IFN-γ-producing tumor-infiltrated CTLs (Supplementary Fig. 9) were markedly increased by the combined treatment of anti-PD-L1 Abs and FimH (Fig. 7b). The IFN-γ production, in response to AH1A5, was also significantly increased in tumor-drLN cells (Fig. 7c, d). Thus, these data suggest that the FimH adjuvant enhanced the effects of anti-PD-L1 Abs in the treatment of cancer.

**FimH induces human peripheral blood DC (PBDC) activation.** FimH effectively induced DC activation in mice; therefore, we next examined its effect on human peripheral blood DCs (PBDCs) to determine whether it could be used as an adjuvant for humans. BDCA1$^+$ and BDCA3$^+$ cells from live lineage$^-$ CD11c$^+$ cells in human peripheral blood mononuclear cells (PBMCs) were classified as human PBDCs, and the activation of these cells by FimH treatment was analyzed (Fig. 8a). The costimulatory molecules and MHC class I and II levels in both BDCA1$^+$ and BDCA3$^+$ DCs were remarkably upregulated by FimH treatment (Fig. 8b). Consistent with mice in vivo findings, the upregulation of costimulatory molecules and MHC class I and II were dose-dependently increased by FimH (Supplementary Fig. 10). In addition, the expression levels of the costimulatory molecules were time-dependently increased and maintained 36 h after treatment of FimH, while those levels were rapidly decreased by LPS treatment 36 h after treatment (Supplementary Fig. 11). FimH-stimulated BDCA1$^+$ cells also substantially increased proliferation of syngeneic CD4 T cells and IFN-γ-producing CD4 T cells, which were considerably greater than the LPS-induced increases in T-cell activation (Fig. 8c). Thus, these data suggest that FimH can induce human PBDC and T-cell activation.

We also examined whether FimH-induced activation of PBDCs is dependent on the MD2 pathway. The PBMCs were pre-treated with 20 μM MD2 inhibitors for 30 min and subsequently treated with FimH and LPS. The LPS-induced upregulation of costimulatory molecules and MHC class I and II in BDCA1$^+$ and BDCA3$^+$ DCs were significantly inhibited by MD2 inhibitors, whereas, FimH treatment did not decrease the levels of those molecules by the MD2 inhibitors (Fig. 8d). Therefore, these data suggested that the FimH-induced activation of PBDCs does not require MD2.

**Discussion**

As cancer antigens are poorly immunogenic, additional treatment with an adjuvant is desired to enhance Ag-specific immune responses for treatment of cancer[8,16]. Treatment with an adjuvant induces the activation of APCs such as DCs and macrophages[8,16]. Combined treatment of a cancer Ag and an adjuvant greatly elevates Ag-specific immune responses in order to kill the Ag-expressing target cells[16,18,21]. In the present study, we found that the treatment of FimH–induced maturation of mouse DCs in vivo and human DCs ex vivo. Moreover, combined treatments of cancer antigens and FimH promoted Ag-specific immune responses, which consequently inhibited tumor growth in mice in vivo.

For immunotherapy against cancer, CTL activation is essential, as CTLs effectively kill cancer cells[16,26]. In vitro-generated DCs contain different subsets from in vivo DCs, which process Ag presentation and T-cell activation differently. CD8α$^+$ DCs in mice and BDCA3$^+$ DCs in humans have specialized in cross-presenting Ag to the CD8 T cells and also promote Ag-specific CTL activation. In contrast, CD8α$^-$ DCs in mice and BDCA1$^+$ DCs in humans promote CD4 T-cell activation by the direct presentation of Ag[20,27–29]. Although the CD4 T cells do not directly kill the Ag-expressing cells, they produce high amounts of cytokines, such as IFN-γ, TNF-α, IL-4, and IL-17, which enhance the function of CTLs[30]. Therefore, the targeted activation of DC subsets results in T-cell activation that effectively kills Ag-expressing cells. Consistent with this framework, FimH-induced activation of DC subsets promotes CTL activation, as shown by the specific killing of OVA peptide-loaded LN cells by the combined OVA and FimH treatment. Thus, these data suggest that FimH can function as a stimulatory molecule for both human and mouse DC subsets.

TLR4 is an essential pattern-recognition receptor and responsible for the immune activation of the innate immune system[31]. LPS is the most well-known TLR4 ligand that induces the maturation of DCs and the activation of macrophages and NK cells[32,33]. Although LPS functions as an effective adjuvant for enhancing immune responses, it is an endotoxin, and its use induces serious side effects, such as tissue inflammation, in humans and animals[34]. To reduce the cytotoxic effects of LPS, LPS from *Salmonella minnesota* were converted into a mixture of acylated di-glucosamines, called MPLA, to stimulate immune cells through TLR4 and to act as an adjuvant that enhances vaccine activates[12]. However, MPLA has shown to have a less stimulatory effect on DCs and T cells than it has on LPS, especially in terms of cytotoxic T cell activation and memory T-cell generation[35,36]. In addition, MPLA is not water-soluble and must

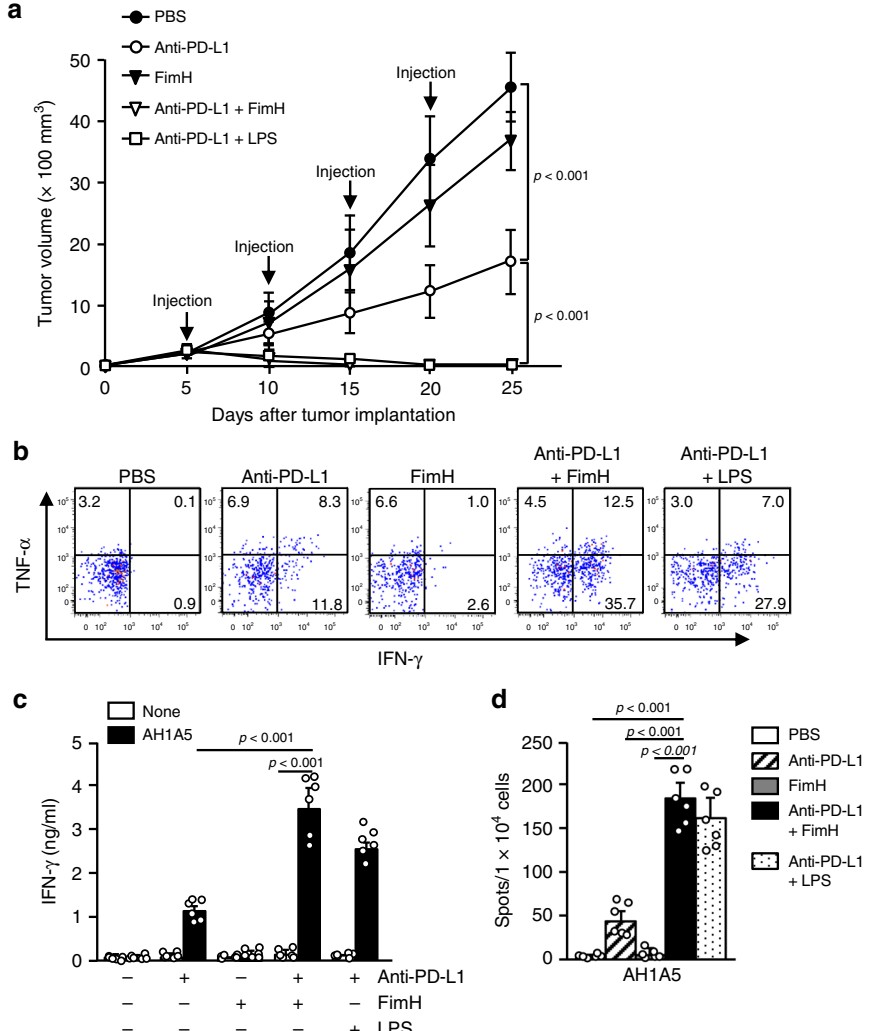

**Fig. 7 Enhancement of the anti-cancer effect of anti-PD-L1 antibodies (Abs) by additional treatment of FimH.** BALB/c mice were injected s.c. with $1 \times 10^6$ CT26 carcinoma cells. Five days after tumor injection, mice were treated i.p. with 10 mg per kg anti-PD-L1 antibodies (Abs), 2.5 mg per kg of FimH, or a combination of anti-PD-L1 Abs and FimH for every 5 days. **a** The graph of CT26 tumor growth is shown. ($n = 6$ mice significance was determined by the log-rank test, mean ± SEM). **b** Intracellular IFN-γ and TNF-α producing CD8 T cells in the tumor are shown. **c** Concentration of IFN-γ in cultured medium from AH1A5 peptide-treated tumor-drLN cells. ($n = 6$ mice, two-way ANOVA, mean ± SEM). **d** AH1A5 peptide-specific IFN-γ production was measured in tumor drLN by ELISpot analysis. ($n = 6$ mice, one-way ANOVA, mean ± SEM).

be included in a formula with trehalose and oil for use in humans and animals[12]. FimH is a water-soluble recombinant protein that also activates DCs via TLR4. More importantly, FimH showed lower cytotoxicity compared to that of LPS in the mouse model. In the current study, we found that FimH time-dependently and dose-dependently elevated the costimulatory molecules expression of DCs in both mouse iLN DCs and human PBDCs, consistent with the previous research[33]. LPS treatment showed a sudden decrease in the level of costimulatory molecules within 24 h, which may be due to the anergy of DCs, but the upregulation effect by FimH was maintained for much longer than the effect induced by LPS. The immune stimulation activity produced by FimH, including DC maturation and T-cell proliferation and cytokine production, is much stronger than that induced by LPS. Furthermore, though the immune adjuvant effect of both LPS and FimH depend on the TLR4 receptor, the recognition of LPS relies on the heterodimer of TLR4 and MD2[37,38], while FimH binds to TLR4 directly independent of MD2[39,40]. In this study, we found that FimH administration induced activation of mouse DCs in vivo and human PMDCs ex vivo in a MD2-independent

manner. In human cells, FimH treatment promotes syngeneic CD4 T cell proliferation and IFN-γ production more efficiently compared to LPS. Given the benefits of FimH, such as water solubility and strong immune-stimulatory effects, it can serve as a adjuvant candidate for the development of vaccines for use in humans and animals.

Compared to other vaccine injection routes, mucosal treatment has advantages, such as reduced risk of blood-borne disease transmission, low costs, and ease in treating children[41]. Due to the limited number of mucosal adjuvants, only a few mucosal vaccines are licensed[41]. CpG oligodeoxynucleotides (ODN), the TLR9 agonists, are potent mucosal adjuvants for nasal vaccination that promote the activation of plasmacytoid DCs (pDCs)[42,43]. However, CpG-ODN have limited immunostimulatory effects in humans because TLR9 is not expressed in human myeloid DCs (mDCs)[44]. In this study, we also found that intranasal treatment with FimH promoted DC maturation in the mLN. While FimH treatment by s.c. injection was found to have a stronger immunostimulatory effect than did LPS treatment, the effect of i.n. treatment with FimH on DC activation and

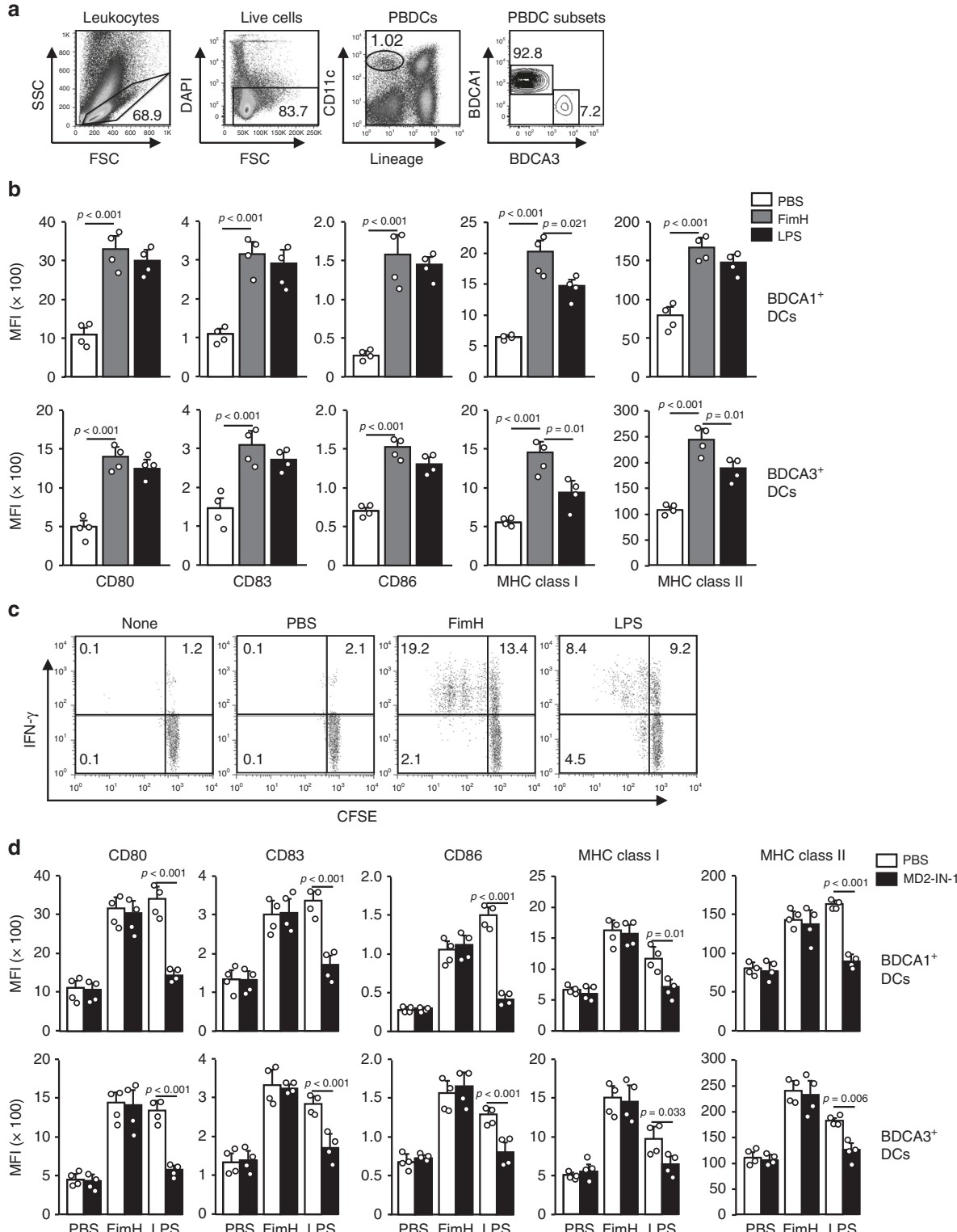

**Fig. 8 FimH induces human peripheral blood DC (PBDC) activation.** Peripheral blood mononuclear cells (PBMCs) were treated with 5 μg per ml of FimH and 2 μg per ml of LPS for 24 h. **a** The distinctness of the PBDCs from the PBMCs is shown. The lineage markers included CD3, CD14, CD16, CD19, CD20, and CD56. **b** The expression levels of costimulatory molecules and MHC classes I and II in BDCA1+ (upper panel) and BDCA3+ DCs (lower panel) are shown. ($n = 4$ donors, one-way ANOVA, mean ± SEM). **c** Syngeneic CD4 T cell proliferation and IFN-γ production were measured after co-cultured with stimulated BDCA1+ DCs and T cells. **d** PBMCs were pre-treated with 20 μM MD2 inhibitors for 30 min and subsequently treated with FimH and LPS. The expression of costimulatory molecules and MHC class I and II in BDCA1+ (upper panel) and BDCA3+ DCs (lower panel) are shown. ($n = 4$ donors, one-way ANOVA, mean ± SEM).

Ag-specific T-cell proliferation in the mLN was no different from that of LPS. Moreover, the prominent role of MD2 in recognition through TLR4 causes the hyporesponsiveness of LPS in the intestine[45], which limited the usage of LPS in the mucosal area. Thus, the direct binding approach of FimH makes it more suitable as a mucosal adjuvant candidate. The fact that the effect of i. n. injection on immune stimulation was weaker than that of s.c. injection may be attributed to the attachment of FimH to the mucosal epithelial cells. FimH is well-known as an adherent molecule in *E. coli*, which binds to intestinal epithelial cells[24]. Therefore, parts of FimH may be lost when it reaches mucosal DC during i.n. treatment, resulting in low DC activation. Although the effect of i.n. treatment on DC activation and Ag-specific T-cell proliferation was similar to that induced by LPS, the combined treatment of FimH and Ag effectively inhibited the growth of lung melanoma with low levels of inflammation.

Our results demonstrate that FimH is a useful adjuvant for the induction of Ag-specific immune responses. These responses promote the inhibition of Ag-expressing tumor growth, including that of B16 melanoma and CT26 carcinoma cells in mice in vivo. Additionally, it can also promote human PBDC and T-cell activation. FimH is thus a promising candidate for the development of an immunotherapeutic vaccine adjuvant for use against cancer in humans.

## Methods

**Mice and cell lines**. C57BL/6, TLR2-KO, TLR4-KO, OT-I, OT-II TCR transgenic mice, C57BL/6-Ly5.1 (CD45.1) congenic mice, and BALB/c mice were obtained from Shanghai Public Health Clinical Center (SPHCC), MD2-KO mice (B6.129P2-Ly96 KO) on C57BL/6 background were kindly provided by Professor Guang Liang (Wenzhou Medical University, Wenzhou, Zhejiang, China) and kept under pathogen-free conditions. The mice were housed in a room at 20–22 °C and 50–60% humidity, and fed standard rodent chow and water. The study was carried out using the guidelines of the Institutional Animal Care and Use Committee at SPHCC. The mouse protocol was approved by the Ethics of Animal Experiments Committee of SPHCC is 2018-A049-01. $CO_2$ inhalation euthanasia was used to euthanize the mice. The murine carcinoma cell line CT26 (ATCC, CRL-2638), murine melanoma cell line B16F10 (ATCC, CRL-6475), and OVA-expressing B16F10 (B16-OVA) were cultured in RPMI 1640 (included 2 mM glutamine, 100 μg per ml streptomycin, 10% FBS, 100 U per ml penicillin, and 1 M HEPES). All cell lines were cultured at 37 °C in a humidified atmosphere of 5% $CO_2$ and air.

**Ethics statement**. This study has been conducted according to the principles of the Declaration of Helsinki. Peripheral blood samples were harvested from healthy donors at the SPHCC. Written informed consent was obtained from all volunteers. The study is approved by the Institutional Review Board at SHAPHC Ethics Committee (IRB number: 2017-Y037).

**Purification of the recombinant FimH protein**. Competent cells were mixed with pET28a-FimH plasmids and maintained in ice for 30 min. The cells were incubated in a 42 °C water bath for 60–90 s and then quickly transferred to ice for two to three minutes. After incubation in LB medium at 37 °C for 1 h with shaking at $0.94 \times g$ (formula: g = r × 11.18 × $10^6$ × rpm², amplitude of vibration for shaking incubator = 2.6 cm), the competent cells were plated onto LB plates containing kanamycin and incubated overnight at 37 °C. A single colony was transferred to one liter of the culture medium with kanamycin and incubated at 37 °C with shaking at $1.41 \times g$. After the cells were harvested by centrifugation on the following day, they were disrupted by ultrasonication at a power of 60% for two seconds each at 2-s intervals for 15 min. The supernatants of the cells were collected by centrifugation at $10,000 \times g$ for 15 min. Using a low-pressure chromatography system, a Ni column was equilibrated with an equilibration buffer (8 M urea; 100 mM $NaH_2PO_4$; 100 mM Tris, pH 8.0) at a flow rate of 0.5 ml per min. The cell supernatant was loaded onto the column and the equilibration buffer was passed through the column at a flow rate of 0.5 ml per min until the OD280 value reached baseline. The column was then washed with a Ni-IDA wash buffer (8 M urea; 100 mM $NaH_2PO_4$; 100 mM Tris, pH 6.3) at a flow rate of 1 ml per min until the OD280 value of the effluent reached baseline. The target FimH protein was eluted with a Ni-IDA elution buffer (8 M urea; 100 mM $NaH_2PO_4$; 100 mM Tris, pH 4.5) at a 1 ml/min flow rate, and the effluent was collected (Supplementary Fig. 1). For production of FimH from the yeast, full-length splicing primers were constructed with protective bases at both ends. A synthetic gene, FimH, was ligated into the expression vector pPIC9K through the cloning sites EcoR I and Not I. Later, the recombinant plasmid pPIC9K-FimH was transferred into a TOP10 clone strain. After enzymatic digestion and sequencing, the plasmid was extracted, and

the amount of the plasmid extracted was more than 10 μg. The Sac I linearized recombinant plasmid pPIC9K-FimH was then electro-transformed into Pichia pastoris GS115, and 11 clones were selected and verified by PCR. Finally, one positive strain was selected for protein expression (Supplementary Fig. 6a). The purified FimH protein was then renatured using a 14-kd dialysis bag in a refolding buffer with 2 M of urea for 4 h and 1 M of urea for another 4 h. The dialysis bag was further placed in a refolding buffer overnight with 0.5 M of urea and additionally placed twice in PBS for 4 h. Finally, the FimH protein was treated with endotoxin removal resin according to the manufacturer's instructions (Detoxi-gel: Thermo Fisher Scientific, Waltham, MA, USA).

**Antibodies (Abs)**. For mouse Abs: isotype control Abs (IgG1, IgG2a or IgG2b), CD11c-APC-Cy7 (N418, #117324, 1:40), CD8α-PerCP-Cy5.5(53-6.7, #100734, 1:50), CD40-Alexa Fluor® 647 (HM40-3, #102911, 1:40), CD80-PE (16-10A1, #104707, 1:20), CD86-PE-Cy7 (GL-1, #105013, 1:40), anti-MHC class I-Alexa Fluor® 647 (AF6-88.5.3, #116511, 1:40), anti-MHC class II-Pacific Blue (M5/114.15.2, #107620, 1:40), anti-IFN-γ-PE-Cy7 (XMG1.2, #505825, 1:40), and anti-TNF-α- PE-Cy7 (MP6-XT22, #506323, 1:40) were procured from BioLegend; anti-PD-L1 (B7-H1, #BE0101) were obtained from BioXcell (West Lebanon, NH, US). For the human Abs: isotype control Abs (IgG1, IgG2a or IgG2b), CD11c (3.9, #254813, 1:25), CD80 (2D10, #207831, 1:20), CD83 (HB15e, #147674, 1:40), CD86 (BU63, #202906, 1:40), anti-HLA-A, B, C (W6/32, #212641, 1:40), anti-HLA-DR, DP, DQ (Tü39, #211013, 1:40), anti-CD1c (L161, #331510, 1:40), anti-CD141 (M80, #344112, 1:40), and anti-IFN-γ (4S.B3, #193274, 1:40) were obtained from BioLegend (San Diego, CA, US).

**Flow cytometry analysis**. Cells were incubated with the unlabeled isotype control Abs and Fc-block Abs for 15 min (BioLegend, San Diego, CA, USA). Later, fluorescence-conjugated Abs had been added, and the cells were further incubated on ice for 30 min. After washing with PBS, the cells were analyzed on FACS Fortessa (Becton Dickinson, Franklin Lakes, New Jersey, US) using FlowJo 8.6 software (Tree Star, San Diego, CA, US). Cellular debris and dead cells were excluded by forward- and side-scatter gating and 4′,6-diamidino-2-phenylindole (DAPI) (Sigma-Aldrich, St. Louis, Missouri, US) staining.

**Mouse lymphoid DC analysis**. iLN and mLN were homogenized and digested by collagenase for 20 min at room temperature. Cells were washed and the pellets were re-suspended in 5 mL Histopaque-1077 (Sigma-Aldrich, St. Louis, Missouri, US) medium. An additional 5 ml of the medium was added on the upper layer of the cell suspension and 1 ml FBS was layered above. The cells were centrifuged at $1700 \times g$ for 10 min. The leukocyte fraction (<1.077 g per cm³) was harvested and stained with fluorescence-labeled monoclonal Abs (mAbs) for 30 min. Anti-B220-FITC (RA3-6B2, #103205, 1:40), anti-CD3-FITC (17A2, #100203, 1:40), anti-CD49b-FITC (DX5, #108905, 1:40), anti-Gr1-FITC (RB68C5, #108405, 1:40), anti-Thy1.1-FITC (OX-7, #202503, 1:40), and anti-TER-119-FITC (TER-119, #116205, 1:40) were added as lineage markers. Lymphoid DCs were defined as a lineage⁻CD11c⁺ cells, of which it populations were further separated and represented as CD8α⁺ and CD8α⁻ DCs.

**Antigen peptides**. OVA peptide 257–264 (SIINFEKL), OVA 323–339 (ISQAV-HAAHAEINEAGR), TRP2 peptide 180-188 (SVYDFFVWL), and AH1A5 peptide (SPSYAYHQF) were purchased from Phtdpeptides Co. Ltd (Henan, China).

**Intracellular cytokine staining**. The spleen and tumors were digested with a digestion buffer (2% FBS and collagenase IV). The splenocytes ($1 \times 10^6$) and tumor cells ($1 \times 10^6$) were stimulated in vitro for 4 h with phorbol 12-myristate 13-acetate (PMA; 50 ng per ml) and ionomycin (1 μM; both from Merck, Kenilworth, New Jersey, USA), with the addition of monensin solution (BioLegend, San Diego, California, USA) during the final 2 h. For intracellular cytokine staining, cells were initially stained to assess surface molecules in the dark for 20 min at room temperature, then fixed and permeabilized with Cytofix/Cytoperm buffer (eBioscience, San Diego, California, USA) for 20 min at room temperature and subsequently incubated with anti-cytokine Abs in Perm/Wash buffer (eBioscience, San Diego, California, USA) for 30 min in the dark at room temperature. Staining of isotype control IgGs was performed in all experiments.

**ELISA**. IFN-γ, IL-1β, IL-6, IL-12p40, and TNF-α concentrations in the serum were measured in triplicates using ELISA kits according to the manufacturer's instructions (Biolegend, San Diego, California, USA).

**Real-time PCR**. Total RNA was extracted from cells and reverse-transcribed into cDNA by Oligo (dT) and M-MLV reverse transcriptase (Promega, Madison, Wisconsin, USA). The cDNA was subjected to real-time PCR (Qiagen, Hilden, Germany) for 40 cycles with an annealing and extension temperature of 60 °C, on a LightCycler 480 Real-Time PCR System (Roche, Basel, Switzerland). Primer sequences are: mouse β-actin forward, 5′-TGGATGACGATATCGCTGCG-3′, reverse, 5′-AGGGTCAGGATACCTCTCTT-3′; T-bet forward, 5′-CAACAAC CCCTTTGCCAAAG-3′, reverse, 5′-TCCCCCAAGCATTGACAGT-3′; GATA3 forward, 5′-CGGGTTCGGATGTAAGTCGAGG-3′, reverse, 5′-GATGTCCCTG

CTCTCCTTGCTG-3′; RORγt forward, 5′-CCGCTGAGAGGGCTTCAC-3′, reverse 5′-TGCAGGAGTAGGCCACATTACA-3′; IFN-γ forward, 5′-GGAT GCATTCATGAGTATTGC-3′, reverse, 5′-CTTTTCCGCTTCCTGAGG-3′; IL-4 forward, 5′-ACAGGAGAAGGGACGCCAT-3′, reverse, 5′-GAAGCCCTACAGAC GAGCTCA-3′; and IL-17A forward, 5′-GCGCAAAAGTGAGCTCCAGA-3′, reverse, 5′-ACAGAGGGATATCTATCAGGG-3′.

**OT-I and OT-II T cell proliferation.** Using cell-isolation kits (Miltenyi Biotec, Bergisch Gladbach, Germany), CD4 T cells and CD8 T cells were purified from OT II and OT I mice, respectively. The cells were suspended in PBS/0.1% BSA containing 10 μM Carboxyfluorescein succinimidyl ester (CFSE) (Invitrogen, San Diego, California, USA) and incubated for 10 min. CFSE-labeled cells ($1 \times 10^6$) were injected i.v. into CD45.1 congenic mice, and 24 h later, mice were injected with either PBS alone, or 2.5 mg per kg of OVA in PBS, or 2.5 mg per kg FimH, or a combination of FimH and OVA in PBS. At 72 h after treatment, tumor and tumor-drLN were harvested and cell proliferation of OT-I or OT-II T was determined by analyzing the CFSE fluorescence intensity using flow cytometry (FACS Fortessa, Becton Dickinson, Franklin Lakes, New Jersey, US) and FlowJo 8.6 software (Tree Star, San Diego, CA, USA).

**In vivo cytotoxicity assay.** Mice were injected i.v. with either a mixture of splenocytes that were differentially labeled with CFSE (200 nM) and loaded with 100 nM SIINFEKL peptide, or with spleen cells labeled with 10 mM CellTracker™ Orange CMTMR (Life technologies), but without loading of the peptide. A total of $10 \times 10^6$ cells per mouse were injected, consisting of a mixture of each target cell population. Splenocytes were collected 24 h after injection of the target cells. Percentage killing was calculated using the formula, as described by FACS Fortessa (Becton Dickinson, Franklin Lakes, New Jersey, USA).

**ELISPOT assay.** Mouse IFN-γ ELISPOT was performed according to the manufacturer's protocol (eBioscience, San Diego, California, USA). In short, LNs were harvested from treated mice and mononuclear cells were isolated by density-gradient centrifugation. The cells were seeded at $50 \times 10^3$ cells per well in a pre-coated plate. The cells then were stimulated with 2 μg per mL of Ag peptides or a negative control peptide at 37 °C for 24 h. Spots in the ELISPOT plates were counted automatically using a CTL ELISPOT reader (CTL Europe GmbH, Bonn, Germany), and the number of spots observed with the control peptide was subtracted from the number of spots with specific peptides for each mouse.

**Hematoxylin and eosin staining.** Lung were infused with 1 ml of 4% paraformaldehyde solution and harvested. The lung samples were fixed in 4% paraformaldehyde solution overnight at 4 °C. The fixed lung samples were embedded in paraffin, and sectioned to 5 μm thickness from different areas across the lung. The sections were placed on slide glasses, de-paraffinized and hydrated, and stained with hematoxylin and eosin (H&E) and examined for tissue damage under a microscope.

**Human PBDC analysis.** The PBMCs were isolated from whole blood by a density-gradient centrifugation method using Ficoll Histopaque and then stained with FITC-conjugated lineage Abs, anti-CD3 (HIT3a, #300306, 1:40), anti-CD14 (63D3, #367115, 1:40), anti-CD16 (3G8, #302006, 1:40), anti-CD19 (HIB19, #302206, 1:40), anti-CD20 (2H7, #302304, 1:40), and anti-CD56 (5.1H11, #362545, 1:40). The lineage⁻CD11c⁺ cells were gated as PBDCs using flow cytometry. The PBDCs were further fractionated into BDCA1⁺ (L161, #331510, 1:40) and BDCA3⁺ (M80, #344112, 1:40) DC cells by FACS Fortessa (Becton Dickinson, Franklin Lakes, New Jersey, USA).

**Syngeneic T cell proliferation assay.** CD1c DC isolation kit (Miltenyi Biotec, Bergisch Gladbach, Germany) was used for isolation of BDCA1⁺ DCs from PBMCs and the isolated cells were stimulated with FimH or LPS for 24 h. Syngeneic CD4 T cells were also isolated from PBMCs of the same donor and labeled with CFSE. The BDCA1⁺ DCs and CD4 T cells were co-cultured with a ratio of 1:25 for 3 days; CFSE staining and intracellular IFN-γ production in CD4⁺ cells were analyzed by flow cytometry (Becton Dickinson, Franklin Lakes, New Jersey, USA).

**Statistical analysis.** All data are expressed as the mean ± standard error of the mean (SEM). A one- or two-way ANOVA (Tukey multiple comparison test) and the Mann–Whitney t-test were used for analysis of the data sets. P-values < 0.05 were considered as statistically significant.

**Reporting summary.** Further information on research design is available in the Nature Research Reporting Summary linked to this article.

## Data availability

All relevant data available in the article, Supplementary Information or from the corresponding author upon request. All the source raw data underlying Figs. 1, 2a–c, e, 3a, b, d, e, 4c, d, 5e, f, 6a, c–e, 7c, d and 8b, d and Supplementary Fig. 5 are provided as a Source data.

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

## Acknowledgements

We thank the SPHCC animal facility for maintaining the animals utilized in this study. This study was supported by Research Fund for the National Research Foundation of Korea (NRF-2019R1C1C1003334) and the National Natural Science Foundation of China (81874164).

## Author contributions

J.O.J. designed the experiments and analyzed data, and J.O.J. and P.L. wrote the paper. M.K., P.L., G.L., X.Z., and J.X. reviewed the paper. W.Z., L.X., H.B.P., J.H., and J.O.J. performed experiments and analyzed data.

## Competing interests

The authors declare no competing interests.
