## [Peer Review File · Nature Communications]

Reviewers' comments:

Reviewer #1 (Remarks to the Author):

IN the present manuscript, the authors identify a novel TLR4 agonist derived from E. coli, FimH, and demonstrate its use as a vaccine adjuvant. Examining both in vitro and in vivo activity, they show that, much like LPS, FimH induces dendritic cells and downstream T cell activation, and can facilitate immunity against tumors in standard mouse orthotopic cancer models. Finally they show that in vitro, FimH can augment human T cell proliferation in a manner similar to LPS.

While the experiments are well performed and the data appear to be reliable, there is little in the way of novelty to recommend the manuscript. Other than having identified a novel TLR4 agonist, there is nothing new shown other than exactly what might need to be demonstrated to prove that FimH can indeed act as a vaccine adjuvant. However, this adjuvant activity is no better than LPS, a molecule not able to be used in any real clinically relevant setting. Is FimH less toxic? Is it like MPL in that it engages internal TLR4 signaling and is less dependent on MyD88? Is there any reason to expect this to be better than other TLR4-targeting adjuvants already in the clinic?

The data in their current form are really only appropriate for a vaccine-oriented journal.

Reviewer #2 (Remarks to the Author):

This is an interesting study on E. Coli-derived FimH as a TLR4 agonist in anti-cancer vaccination.

The experiments are well-performed, the data are convincing, and interpreted adequately.

FimH displays TLR4 agonist activity that is very similar to LPS. While FimH is somewhat more potent in many of the assays, this could be due to the doses of both agents, especially since in most experiments more FimH than LPS is used.

It would be important to show superiority of FimH over LPS by some assay. For example, what is the LD50 of both agents, compared to their minimally and maximally effective doses in vivo? Does FimH have a better activity/toxicity profile? If it behaves so similarly to LPS, what is the value of having FimH as a TLR4 agonist (other than its water-solubility)?

By far the biggest, and extremely critical, question this study leaves open is whether FimH is truly a TLR4 agonist. Even though this has been asserted in a previous publication, it needs to be shown without a shadow of a doubt that the activity of FimH is not due to LPS (or another contaminant) from the E. Coli producer cells. All the results in the manuscript could be easily explained by LPS in the FimH preparation. LPS could copurify with FimH on the Ni column used to sort FimH, and the LPS removal resin may fail to completely remove all LPS, especially if it is associated with FimH. It is really critical that the authors prove in multiple ways that the observed results are not due to copurified LPS. Can they express FimH in an LPS-negative eukaryotic cell or bacteria? Or use classic, sensitive LPS assays to prove that LPS is not present in a free or protein-bound form? Without ironclad proof that there is no trace of LPS present, it is difficult to accept that FimH is indeed a TLR4 agonist, and not simply a carrier (and possibly potentiator) of LPS.

Overall, the notion that FimH is a TLR4 agonist that can be used instead of LPS to obtain similar results is interesting. If the authors can provide evidence of that the activity of this preparation is truly due to FimH and not to LPS, the impact of this finding would still seem to be somewhat limited. The study does not provide any new understanding or mechanistic insights beyond what is known for other TLR4 agonists such as LPS, and there is no indication that it is less toxic or more therapeutic. Thus, the scientific or medical impact of this study could be considered modest until FimH is shown to be dramatically superior to LPS (or MPL), and/or is applied clinically.

Minor points:

In figure 4, sometimes TRP-2 is misspelled as TPR-2.

In figure 4C, please quantify the lung nodules.

In figure 4D, please quantify the tumor surface per lung.

We revised the previous version of the manuscript, and we ask that our revised manuscript be re-evaluated for publication. We made revisions based on the referee reports (NCOMMS-17-32533), for which we have provided responses below. We performed several experiments, added two new figures, altered two figures to include appropriate controls, and revised the corresponding text for clarity.

Reviewers' comments:

Reviewer #1 (Remarks to the Author):

IN the present manuscript, the authors identify a novel TLR4 agonist derived from *E. coli*, FimH, and demonstrate its use as a vaccine adjuvant. Examining both in vitro and in vivo activity, they show that, much like LPS, FimH induces dendritic cells and downstream T cell activation, and can facilitate immunity against tumors in standard mouse orthotopic cancer models. Finally they show that in vitro, FimH can augment human T cell proliferation in a manner similar to LPS.

While the experiments are well performed and the data appear to be reliable, there is little in the way of novelty to recommend the manuscript. Other than having identified a novel TLR4 agonist, there is nothing new shown other than exactly what might need to be demonstrated to prove that FimH can indeed act as a vaccine adjuvant. However, this adjuvant activity is no better than LPS, a molecule not able to be used in any real clinically relevant setting. Is FimH less toxic? Is it like MPL in that it engages internal TLR4 signaling and is less dependent on MyD88? Is there any reason to expect this to be better than other TLR4-targeting adjuvants already in the clinic?

Answer: We thank the reviewer for these comments. We conducted further studies to demonstrate that FimH can be used as an adjuvant for cancer therapy in humans. As the reviewer mentioned, we compared the effects of FimH and LPS in inducing inflammation and toxicity in mice and found that FimH (250 mg/kg) promoted much lower levels of proinflammatory cytokine production and lung inflammation compared to those induced by LPS. In addition, the lethal dose of FimH was much higher than that of LPS; the lethal dose of FimH was 100-fold higher than the working concentration. Therefore, these data indicate that FimH has much lower cytotoxicity compared to LPS.

We also evaluated the efficacy of FimH as an adjuvant for the anti-cancer effects of the anti-PD-L1 antibody and found that FimH enhanced the anti-PD-L1 antibody-mediated anti-cancer effects in mice.

Taken together, these data suggest that FimH may be used as a vaccine adjuvant for humans due to its low toxicity and high immunostimulatory function.

The data in their current form are really only appropriate for a vaccine-oriented journal.

Reviewer #2 (Remarks to the Author):

This is an interesting study on *E. Coli*-derived FimH as a TLR4 agonist in anti-cancer vaccination.

The experiments are well-performed, the data are convincing, and interpreted adequately.

FimH displays TLR4 agonist activity that is very similar to LPS. While FimH is somewhat more potent in many of the assays, this could be due to the doses of both agents, especially since in most experiments more FimH than LPS is used.

It would be important to show superiority of FimH over LPS by some assay. For example, what is the LD50 of both agents, compared to their minimally and maximally effective doses in vivo? Does FimH have a better activity/toxicity profile? If it behaves so similarly to LPS, what is the value of having FimH as a TLR4 agonist (other than its water-solubility)?

Answer: We thank the reviewer for these crucial comments. As the reviewer mentioned, we examined the lethal concentration of FimH for mice and found that 250 mg/kg of FimH treatment induced 30% of mice death. However, 20 mg/kg of LPS-treated mice have all died within 36 hours after injection. Moreover, the high concentration of FimH (250 mg/kg) treatment showed much lower levels of lung inflammation compared to 20 mg/ml of LPS treatment. The lethal concentration of FimH was 100 times higher than working doses of FimH but lethal concentration of LPS was only 20 times higher than working dose. These data suggest that FimH may be useful as a vaccine adjuvant in humans due to its low toxicity.

By far the biggest, and extremely critical, question this study leaves open is whether FimH is truly a TLR4 agonist. Even though this has been asserted in a previous publication, it needs to be shown without a shadow of a doubt that the activity of FimH is not due to LPS (or another contaminant) from the *E. Coli* producer cells. All the results in the manuscript could be easily explained by LPS in the FimH preparation. LPS could copurify with FimH on the Ni column used to sort FimH, and the LPS removal resin may fail to completely remove all LPS, especially if it is associated with FimH. It is really critical that the authors prove in multiple ways that the observed results are not due to copurified LPS. Can they express FimH in an LPS-negative eukaryotic cell or bacteria? Or use classic, sensitive LPS assays to prove that LPS is not present in a free or protein-bound form? Without ironclad proof that there is no trace of LPS present, it is difficult to accept that FimH is indeed a TLR4 agonist, and not simply a carrier (and possibly potentiator) of LPS.

Answer: Thank you for your feedback on important factors. For ruling out the possibility of contamination of LPS in FimH, we extracted FimH from yeast, as we have shown in Figure 4. FimH extracted from yeast showed similar DC activation ability with that of FimH purified by *E. coli*. We also found that FimH treatment in bone marrow-derived DCs (BMDCs) did not promote IL-1 β production, whereas LPS treatment induced marked upregulation. Therefore, the pattern of DC activation by FimH is different from that induced by LPS.

Overall, the notion that FimH is a TLR4 agonist that can be used instead of LPS to obtain similar results is interesting. If the authors can provide evidence of that the activity of this preparation is truly due to FimH and not to LPS, the impact of this finding would still seem to somewhat limited. The study does not provide any new understanding or mechanistic insights beyond what is known for other TLR4 agonists such as LPS, and there is no indication that is less toxic or more therapeutic. Thus, the scientific or medical impact of this study could be considered modest until FimH is shown to be dramatically superior to LPS (or MPL), and/or is applied clinically.

Answer: As we have mentioned above, we showed that the FimH-induced immune stimulation pattern is different from those induced by LPS. Moreover, yeast-derived FimH has the same function as *E. coli*-derived FimH in DC activation in mice in vivo. Therefore, contaminated LPS cannot be the main contributor to DC

activation in FimH. In addition, we further evaluated the efficacy of FimH as an adjuvant for the anti-PD-L1 antibody, which is actively used in clinical practice for cancer immunotherapy. As we have shown in Figure 7, FimH elevated the anti-cancer effect of the anti-PD-L1 antibody. Therefore, the low toxicity of FimH may make it useful as adjuvant immunotherapy for human cancer.

Minor points:

In figure 4, sometimes TRP-2 is misspelled as TPR-2.

Answer: We apologize for the errors. We have corrected them.

In figure 4C, please quantify the lung nodules.

Answer: We have added a graph of the lung nodule numbers.

In figure 4D, please quantify the tumor surface per lung.

Answer: We have added a graph of metastatic cancer lesion in lung.

Reviewers' comments:

Reviewer #1 (Remarks to the Author):

The authors have added some important data in regards to the toxicity profile of the FimH relative to LPS, and their titration clearly shows a better tolerance of the FimH (Fig 2). However, once again, they are left with very few differences between the use of FimH and LPS and a substantial number of their data sets (Figures 4-7). their minor differences seen in Figure 3 could very well be just a dose issue and unless a titration of LPS and FimH are done, real differences cannot be well concluded, especially given all of the in vivo data where there are no differences. this is especially true of the human cellular studies... there is no dose curve for the parameters analyzed, and so no way of knowing whether a different dose of LPS would produce the same or better results.

As a side note, a major point that needs to be corrected is the assessment of proliferating T cells using the CFSE gating they have shown (Fig 3). it is not acceptable to simply gate on all cells with diluted CFSE and report that percentage as the percent divided. this is because the number of dividing cells are the result not only of division but also of survival, such that a starting population of 100 cells in which only 10% began dividing would result in 1280 cells after 7 division. if gated as shown in figure 3, one would aberrantly conclude that 93% of the cells had undergone cell division (90/1280). to calculate this correctly, the authors need to use the FloJo module specific for this purpose, based on a paper by Mario Roederer (Roederer, M. Interpretation of cellular proliferation data: avoid the panglossian. Cytometry A 79, 95-101 (2013))

Reviewer #3 (Remarks to the Author):

The authors have adequately addressed this reviewer's questions. As previously mentioned, the true novelty of the findings remains questionable, given the discovery of FimH as a TLR4 agonist in 2008. The current study elegantly and convincingly demonstrates that FimH can be applied in cancer immunotherapy, as would be expected of a TLR4 agonist. It appears less toxic than LPS, but this is in mice, and its toxicity profile (a main possible advantage of FimH) in human remains unknown. Given this, and the lack of fundamentally new biological mechanism, it is difficult for me to recommend acceptance for publication in Nature Communications.

Reviewer #1 (Remarks to the Author):

The authors have added some important data in regards to the toxicity profile of the FimH relative to LPS, and their titration clearly shows a better tolerance of the FimH (Fig 2). However, once again, they are left with very few differences between the use of FimH and LPS and a substantial number of their data sets (Figures 4-7). their minor differences seen in Figure 3 could very well be just a dose issue and unless a titration of LPS and FimH are done, real differences cannot be well concluded, especially given all of the in vivo data where there are no differences. this is especially true of the human cellular studies... there is no dose curve for the parameters analyzed, and so no way of knowing whether a different dose of LPS would produce the same or better results.

Answer (A): We appreciate the reviewer's important comments. In response, we further investigated the differences between LPS and FimH and revised the manuscript. We examined time- and dose-dependent effects of FimH and LPS in human PBDC and mouse DC activation. Importantly, the pattern of DC activation by FimH and LPS was quite different depending on the time and dose, as shown in Figures S3, S4, S7, and S8. In addition, we examined the differential effects of FimH compared with LPS in TLR4 stimulation, in which LPS-induced activation of DCs required MD2 but FimH was MD2-independent. LPS did not promote activation of DCs in MD2-knockout mice and by inhibition of MD2, while FimH treatment still induced activation of those DCs. The less cytotoxic effect of FimH compared to LPS may be due to the MD2-independent stimulation of DCs, as the prominent role of MD2 is promoting the hyporesponsiveness of LPS by recognition through TLR4 (J Immunol, 176 (7), 4258-66 (2006)).

As a side note, a major point that needs to be corrected is the assessment of proliferating T cells using the CFSE gating they have shown (Fig 3). it is not acceptable to simply gate on all cells with diluted CFSE and report that percentage as the percent divided. this is because the number of dividing cells are the result not only of division but also of survival, such that a starting population of 100 cells in which only 10% began dividing would result in 1280 cells after 7 division. if gated as shown in figure 3, one would aberrantly conclude that 93% of the cells had undergone cell division (90/1280). to calculate this correctly, the authors need to use the FloJo module specific for this purpose, based on a paper by Mario Roederer (Roederer, M. Interpretation of cellular proliferation data: avoid the panglossian. Cytometry A 79, 95-101 (2013))

A: We thank the reviewer for the crucial comments. We analyzed the T cell proliferation data again based on the reviewer's recommendation.

Reviewer #3 (Remarks to the Author):

The authors have adequately addressed this reviewer's questions. As previously mentioned, the true novelty of the findings remains questionable, given the discovery of FimH as a TLR4 agonist in 2008. The current study elegantly and convincingly demonstrates that FimH can be applied in cancer immunotherapy, as would be expected of a TLR4 agonist. It appears less toxic than LPS, but this is in mice, and its toxicity profile (a main possible advantage of FimH) in human remains unknown. Given this, and the lack of fundamentally new biological mechanism, it is difficult for me to recommend acceptance for publication in Nature Communications.

Answer (A): We thank the reviewer for these comments. We agree that FimH was discovered to be a TLR4 agonist long ago. However, the adjuvant effect of FimH for cancer immunotherapy has not been investigated. As the reviewer knows, the cytotoxic effect of a novel reagent in humans cannot be evaluated. Instead, we further examined the difference between FimH and LPS in stimulation of TLR4 and found that FimH-induced stimulation of TLR4 was independent of MD2. MD2 is also known to be a cofactor for LPS binding to TLR4 and to contribute to the hyporesponsiveness of LPS. We found that FimH-induced activation of DCs was independent in mouse DC *in vivo* and human PBDC *ex vivo*, which may be the reason that FimH has less toxicity than LPS. Thus, the FimH can be a candidate adjuvant for enhancing Ag-specific immunity and immune check point blockade activity for treatment of cancer.

REVIEWERS' COMMENTS:

Reviewer #1 (Remarks to the Author):

The authors have addressed the majority of the technical concerns of previous reviews. While they redid the analysis of the CFSE-based calculations on percent divided, they failed to modify the graphs in figure 3A which still show ~90% division, a number that is certainly incorrect.

There is not any more to add to this manuscript to improve its current point. The issue is not one of technical competency, it continues to be one of novelty... the reviewers have broadly and repeatedly questioned the novelty and impact of the data. This questionable novelty still stands. In terms of in vivo anti-tumor effect, LPS is as good if not better (Figure 7) than FimH. Whatever comparisons they make in vitro to see differences, these appear to be irrelevant in vivo. The fact that another TLR4 agonist can help serve as an adjuvant for anti-tumor activity isn't of sufficient novelty and impact to warrant a Nat Comm paper in my, and apparently the other reviewers, point of view.

Reviewer #1 (Remarks to the Author):

The authors have addressed the majority of the technical concerns of previous reviews. while they redid the analysis of the CFSE-based calculations on percent divided, they failed to modify the graphs in figure 3A which still show ~90% division, a number that is certainly incorrect.

Answer (A): We apologize for the error. We have re-analyzed the data, and the revised results are shown in Figure 3a.

There is not any more to add to this manuscript to improve its current point. the issue is not one of technical competency, it continues to be one of novelty... the reviewers have broadly and repeatedly questioned the novelty and impact of the data. this questionable novelty still stands. in terms if in vivo anti-tumor effect, LPS is as good if not better (Figure 7) than FimH. Whatever comparisons they make in vitro to see differences, these appear to be irrelevant in vivo. The fact that another TLR4 agonist can help serve as an adjuvant for anti-tumor activity isn't of sufficient novelty and impact to warrant a Nat Comm paper in my, and apparently the other reviewers, point of view.

A: Thank you for these important comments. We agree with the reviewer's comments that the FimH and LPS made in vitro manifest differences but appear to be irrelevant in vivo. It may be due to the different immune responses in vivo and in vitro and the LPS toxicity. Moreover, the TLR4 ligand is well known to induce anti-tumor activity by an immune-stimulating effect, and the protein-based TLR4 ligand is newly defined. Therefore, we kindly ask once again that our paper is considered for publication in Nature Communications.